# Collective Effects in Ionic Liquid [emim][Tf2N] and Ionic Paramagnetic Nitrate Solutions without Long-Range Structuring

**DOI:** 10.3390/molecules27227829

**Published:** 2022-11-13

**Authors:** Eni Kume, Nicolas Martin, Peter Dunne, Patrick Baroni, Laurence Noirez

**Affiliations:** 1Laboratoire Léon Brillouin (CEA-CNRS), Université Paris-Saclay, CEDEX, 91191 Gif-sur-Yvette, France; 2Institut de Physique et de Chimie des Matériaux de Strasbourg, CNRS-UMR7504, 23 rue du Loess, CEDEX 2 BP 43, 67034 Strasbourg, France

**Keywords:** ionic liquids, mechanical measurements, shear elasticity, neutron scattering

## Abstract

Mesoscopic shear elasticity has been revealed in ordinary liquids both experimentally by reinforcing the liquid/surface interfacial energy and theoretically by nonextensive models. The elastic effects are here examined in the frame of small molecules with strong electrostatic interactions such as room temperature ionic liquids [emim][Tf2N] and nitrate solutions exhibiting paramagnetic properties. We first show that these charged fluids also exhibit a nonzero low-frequency shear elasticity at the submillimeter scale, highlighting their resistance to shear stress. A neutron scattering study completes the dynamic mechanical analysis of the paramagnetic nitrate solution, evidencing that the magnetic properties do not induce the formation of a structure in the solution. We conclude that the elastic correlations contained in liquids usually considered as viscous away from any phase transition contribute in an effective way to collective effects under external stress whether mechanical or magnetic fields.

## 1. Introduction

The microfluidic scale is certainly the scale at which the limits of the hydrodynamic conditions are more noticeable. Introducing nonextensive parameters such as the influence of interfacial interactions or the mesoscopic liquid shear elasticity is meaningful when the scale lowers. A weak but non-negligible mesoscopic shear elasticity has recently been highlighted at low frequency (Hz) in various liquids from the macro- to nanoscale, thus covering a wide dimensional range [1,2,3,4,5,6] interpreted by scale-dependent models [7,8]. The submillimeter shear elasticity is important because it calls into question the flow results from the viscous properties of the liquid; liquids would propagate shear waves at frequencies much lower than the inverse of Maxwell’s relaxation time *τ = η/G,* where η is the viscosity, *τ* the molecular time and *G*, the high-frequency shear modulus, is of the order of GPa [9,10]. Shear elasticity indicates that liquid molecules are long-range correlated; i.e., long-range elastic properties are typically measurable up to the submillimeter scale. The literature is abundant with examples illustrating various long-range correlations, for example, in liquid water via mechanical relaxation methods [11], via Rayleigh scattering [12], or long-range (bulk) electrostatic oscillations measured in ionic liquids [13], but perhaps one of the most spectacular is certainly the suspended water bridge [14,15]. We expect that ionic interaction forces might exacerbate elastic correlations in charged liquids such as ionic liquids or ionic solutions, which are the scope of the present study. These liquids are characterized by strong electrostatic interactions due to charge carriers that can challenge the gravity under a magnetic field gradient in the case of paramagnetic liquids.

Ionic liquids have recently gained a lot of interest because of the important progresses in synthesis that produces new molecules exhibiting liquid properties down to room temperatures. They are generally characterized by a very low vapor tension [16], nonflammability, thermal stability and excellent solvating properties. The understanding of the physical properties of (room temperature) ionic liquids is a very actively investigated field. One of the current debated questions is of how to classify these ionic liquids. Should they be classified as conventional liquids [17] or, rather, glassy-like or liquid crystal materials [18]? In viscous ionic liquids, deviations from a Maxwell behavior are frequently observed at low frequencies [19], indicating a topological network. Similarly, light scattering investigations [20] indicate long-range structures reminiscent of the clusters observed by E.W. Fischer [21]. Neutron diffraction [22] and dynamic measurements seem to indicate an undefined frontier between liquid and solid states in terms of similar dynamic and structural correlations while having a different (Colombian) origin for ionic liquids and liquid crystals [23]. Recent dynamic molecular simulations have highlighted the need to include polarization effects to accurately describe the physical properties of ionic liquids [24,25,26,27]. The polarizability parameter even has an important impact in the presence of solvent [26]. A general consensus seems that long alkyl chains favor the formation of self-assembly in ionic systems, i.e., of a long-range structure that can give rise to collective properties such as lamellar phases and their associated long-range structuring, ferromagnetic or paramagnetic properties [27,28,29,30].

Ionic interactions can indeed provide magnetic properties. Paramagnetic properties have been found in liquid metals (at high temperature), molten salts, microemulsions or even aqueous solutions [28,29,30]. A paramagnetic ionic liquid contains an inorganic magnetic cation and exhibits a positive magnetic susceptibility [31], unlike the wide majority of fluids which are diamagnetic. These liquids, called also magnetic ionic liquids, are typically applied for medical imaging as contrast liquid for diagnostic or therapeutic objectives. Paramagnetic liquids have the faculty to exhibit a strong attraction to a magnetic field gradient that defies capillary forces and gravity [30] and enable to access to the determination of their magnetic susceptibility [31]. Figure 1 illustrates the liquid rising in the bottle while the magnetization is only acting on the paramagnetic ion. In nitrate solutions, the cation (Gd, Tb, Fe, Mn, Cu, etc.) does not dissociate, but the whole solution is drained by the magnetic field gradient, demonstrating that a strong coupling between molecules of water and a paramagnetic complex exists (Figure 1). It is calculated that the entropic energy TΔS = −1.8 (k.J mol^−1^) is approximately four orders of magnitude greater than the magnetic energy (at 1T) of a molar solution [32,33]. Water molecules being diamagnetic, a high degree of connectivity between liquid molecules and paramagnetic cation is required, which is incompatible in terms of short relaxation times (and uncorrelated density fluctuations). This effect brings forth questions the real nature of paramagnetic solutions.

Here, we aim at studying an ionic liquid and a paramagnetic ionic solution without spontaneous long-range ordering. For that, we chose a molten salt with a short alkyl chain, the Ethyl-3-methylimidazolium ([emim][Tf2N]) and Terbium or Gadolinium nitrate solutions without alkyl chain for the paramagnetic liquid. In bulk ionic liquids, several structural studies indicate that only Imidazolium-based ionic liquids with long alkyl chains exhibit structural peaks, interpreted by a smectic-like layering of the alkyl chains [34,35,36]. Triolo et al. [36] reported the emergence of a structural peak within the range of 0.2–0.5 Å^−1^ when the chain length contains more than five carbons. [emim][Tf2N], studied in this paper, is made of a small two-carbon-chain molecule complex. The short alkyl chain and the ion symmetry do not favor the molecular stacking necessary for a self-assembly. The liquid phase is characterized by cation–anion interactions with a first peak of the structure factor at 4.9–5.3 Å [34,35,36]. Thus, a discussion about the existence of a prepeak is also irrelevant here. From a structural point of view, [emim][Tf2N] is a simple liquid.

To probe collective properties in the absence of a long-range molecular structure, we first examine how the chosen ionic liquids respond to a mechanical shear field and identify shear elasticity [2,3,4,5]. The fluid is submitted to an oscillatory shear strain of small finite amplitude in accordance with the conventional method of viscoelastic measurements, but the transmission of the shear stress is optimized using total wetting fluid/surface boundary conditions, i.e., a strong surface energy attracts liquid molecules to the solid surface, reducing the occurrence of contact loss and interfacial slip [37].

We will show that the mesoscopic shear elasticity is present in the case of ionic liquids (molten salt) and solubilized salts of Gadolinium or Terbium, opening the way for another possible understanding of giant collective properties. We complete the study by a structural approach analyzing the neutron scattering experiments of Gd(NO_3_)_3_ and Tb(NO_3_)_3_ solutions to determine if the paramagnetic (collective) effects are coupled with a structure formation. We will show that manifestation of the paramagnetic (collective) properties are exhibited without coupling with an induced structure.

## 2. Materials and Methods

The analysis of the dynamic response to a mechanical shear stress provides information on the elastic or the viscous character of materials and allows quantitative measurements of the viscoelastic character versus frequency and shear strain. The shear stress is generated by placing the sample between two coaxial disks, one fixed, the other one coupled to a motor imposing a rotating sinusoidal motion of variable frequency (*ω*) to the disk. The amplitude of the shear strain *γ*_0_ of amplitude is also variable (strain-imposed mode). The second disk is immobile and coupled to a sensor. It measures the stress transmitted by the sample via the torque (*σ*) transmitted by contact to the disk. Oscillatory motion and torque measurement are provided by a conventional rheometer (Ares II—TA-Instruments). Simultaneously, a 7-digit voltmeter (Keitley; rate: 300 data/s) measures the voltage of the motor imposing the oscillation (input wave associated with the strain amplitude), while another 7-digit voltmeter measures the voltage associated with the sensor (output wave associated to the torque). This setup enables the simultaneous measurement of shear-strain and shear-stress signals and of the dynamic profile using the conventional relationship: *σ*(*ω*) = *G*_0_*.γ*_0_*.sin*(*ω.t* + Δ*ϕ*), where *σ*(*ω*) is the shear stress; *G*_0_, the shear modulus; *γ*_0_, the strain amplitude defined as the ratio of the displacement to the sample gap; and Δ*ϕ*, the phase shift between the input and the output waves. This equation can be also expressed in terms of shear-elastic (*G*′) and viscous (*G*″) moduli: *σ*(*ω*) *= γ*_0_.(*G*′(*ω*)*.sin*(*ω.t*) *+ G*″(*ω*)*.cos*(*ω.t*)), with *G*′ the component in phase with the strain and *G*″ the out-of-phase component. It should be stressed that the formalism in terms of *G*′ and *G*″ supposes that the resulting stress wave keeps the shape of the imposed strain wave (sinusoidal-like). The study of the wave shape is thus complementary.

As indicated in the introduction, the liquid was confined between high-energy surfaces (α-Alumina fixtures) to reach total wetting conditions. Due to surface charges, α-Alumina increases the interaction energy at the fluid/surface boundary and therefore reinforces the shear stress transmission following the protocol described in [2,3,4]. Conventional metallic fixtures made generally of aluminum or stainless steel do not fulfill full wetting boundary conditions.

Neutron scattering measurements were carried at the lab. Léon Brillouin (Orphée reactor) was used to probe the q-scattering range from 10^−2^ Å^−1^ up to 3 Å^−1^. A specific beam line was dedicated, adapting the 2D neutron detector Barotron (which offers a high spatial resolution and the possibility of a wide q-range observation [38]) for coupled magnetic-scattering measurements in the intermediate scattering range (5–100 Å, thus accessing the first structural liquid peak at 2 Å^−1^). Specific sample environments were designed to enable the in situ measurement of the impact of the magnetic field or of a gradient of a magnetic field with a control of the intensity, of the field geometry, an adaptation of the setups as a function of the instrument or the liquid studied. The small-angle neutron spectrometer PA20 [39] was also equipped with a specific magnetic field design to optimize the structural study of the paramagnetic fluids in the low q-range. The selected wavelength and the sample-detector distances were 5 Å and 2 m, respectively. The liquids were placed in 1 mm diameter capillary tubes for wide-angle neutron scattering and in 1 mm thick rectangular Hellma cells adapted for small-angle neutron scattering. Neutron spectra were normalized by an incoherent scatterer (Vanadium for wide-angle scattering and Plexiglas for small-angle scattering).

The ionic liquid, 1-Ethyl-3-methylimidazolium bis(trifluoromethylsulfonyl)imide ([emim][Tf2N]), the Terbium(III) and the Gadolinium(III) nitrate hexahydrates Tb(NO_3_)_3_ · 6H_2_O and Gd(NO_3_)_3_ · 6H_2_O, respectively, were purchased from Aldrich company. The glass transition and the melting temperatures of [emim][Tf2N] are −87 °C and −18 °C, respectively. The experiments were carried out at room temperature and the paramagnetic salts were dissolved in water (deuterated water for neutron experiment), giving rise to an optically transparent solution.

## 3. Results

Figure 2a displays the signals over four periods of oscillation of the input sin wave and of the output shear stress of the ionic liquid [emim][Tf2N] measured at 0.100 mm gap thickness and 2% strain amplitude. The study of the wave shape shows that the shear strain and the stress waves are almost superposed, indicating a nearly instant response of the fluid, i.e., an elastic response to the shear strain.

The frequency dependence of the viscoelastic moduli (Figure 2a bottom) confirms that the ionic liquid exhibits an elastic behavior recognizable by a shear modulus about one decade higher than a viscous modulus. Therefore, in conditions close to the equilibrium state (the applied shear strain here is 2%), the dynamic mechanical response indicates that the true nature of the ionic liquid is that of a solid characterized by a weak elastic modulus of about the Pascal unit.

Figure 2b shows the dependence of the viscoelastic response as a function of the amplitude of the shear strain. This measurement is also essential since it shows that the elastic response is progressively lost by increasing the shear strain. A transition from elastic-like to viscous-like is observed above 20% shear strain; the ionic liquid exhibits a yield strength, beyond which the measurement indicates a strain-induced viscous behavior.

The same dynamic mechanical approach is used for paramagnetic liquids. Figure 3 displays the shear stress exhibited by a 0.1 M solution of Gadolinium(III) nitrate (Gd(NO_3_)_3_), which displays paramagnetic properties at room temperature. We carried out measurements at a relatively large gap thickness (430 µm) to fit with the order of magnitude of the scattering measurements (the mesoscopic shear elasticity decreases as the scale increases [2,7,8]). The dynamic spectrum (Figure 3a) shows that the elastic component is larger than the viscous component, both exhibiting a relative independence with respect to the frequency within a range of 0.4 rad/s to 10 rad/s. The paramagnetic liquid exhibits, in nearly static conditions (1% shear strain), an elastic behavior. This elastic behavior is, however, very fragile. Figure 3b illustrates the evolution of the elastic-like behavior as a function of the strain amplitude. The shear elasticity is progressively lost over 10% shear strain, highlighting the need to stay close to equilibrium conditions (weak shear strain and low thickness) to probe the elastic-like response. Similarly to the ionic liquid [emim][Tf2N], the large-strain plastic behavior makes the viscous regime appear. Since the viscous behavior is obtained at large strain amplitudes, it is the strain-induced product of the initial shear elasticity.

In the last section of this paper, we explore the response of the paramagnetic fluid to a magnetic field viewed from a structural point of view. Paramagnetic solutions exhibit a flow coupling at relatively low magnetic fields that challenges conventional thermodynamic considerations. Neutron scattering is a very powerful method able to access bulk structural properties of length scale lying within 5–5000 Å due to the exceptional penetration length of the neutron radiation. The strong absorption coefficient of Terbium or Gadolinium cations is, however, a difficulty that excludes the study of high concentrations for bulk samples. To lower the contribution of the incoherent background, the paramagnetic salts are dissolved in deuterated water. Figure 4a illustrates, by comparison with deuterated water, the wide-angle neutron intensity scattered by a 0.1 M paramagnetic solution of Tb(NO_3_)_3_. The identification of a microstructure or of changes in the microstructure induced by applying a magnetic field should be manifested in the structure factor, which can be determined by probing the scattering in a q-range around 2 Å^−1^, which covers the interaction distances. The first observation is that the peak of the structure factor of the paramagnetic solution is the one of a liquid, and that it is very weak compared to the one of the deuterated waters and barely comes out of the background noise, even using neutron scattering. Figure 4b compares the scattering of the paramagnetic solution with and without magnetic field (uniform magnetic field of 0.02T). The two scattering curves corresponding to the first peak of the structure factor overlap, making it impossible to identify a magnetic field influence in this Q-range corresponding to the interionic distances. No modification of the diffraction pattern was observed in a magnetic field gradient (tested up to 2T field gradient) or by increasing the concentration to 1 M.

The small-angle study was carried out on a series of different concentrations of solutions of Gd(NO_3_)_3_ solubilized in deuterated water. The sample was placed in a 0.2T magnetic field gradient. Varying the concentration allows one to identify a critical solution concentration for which a coupling with the magnetic forces might order the paramagnetic ions in clusters. This test is shown in Figure 5. No structure could be identified in the in the q range from 0.01 up to 0.1 Å^−1^ within the concentration range up to 0.22 M.

Therefore, on the basis of a wide scattering range analysis covering scales from 0.01 to 2 Å^−1^ and a series of different concentrations, we conclude that the paramagnetic properties are not associated with a detectable structuring in the Gd(NO_3_)_3_ solutions up to 0.22 M or in Tb(NO_3_)_3_ solutions up to concentrations of 1 M. Higher concentrations become difficult to use due to the high-absorption cross-sections of Terbium and Gadolinium (23 barns and 49,700 barns at 1.798 Å, respectively [40]). The solutions exhibit a typical scattering pattern of simple (ideal) liquids. Applying a magnetic field (or a gradient of a magnetic field) also does not induce structural change measurable by scattering methods. This result is coherent with previous observations reporting on a significant local ordering only when the ionic liquid possesses long alkyl chains. Correlatively, without long alkyl chains and without structure, the collective response illustrated by the liquid rise in a magnetic field gradient must find other tracks of interpretation. The absence of atomic separation of the paramagnetic complex from the surrounding water molecules in a magnetic field gradient indicates that the ionic interaction forces between the paramagnetic ion and the water molecules are stronger than the force exerted by magnetic field gradient on the paramagnetic ion. The interactions with water molecules are sufficiently strong to allow the liquid to rise against the gravity by magnetization of the paramagnetic ion. As a result, the liquid sustained in the magnetic field gradient does not exhibit hydrodynamic properties (it does not flow). Magnetically, the liquid behaves like a single body. Interestingly, the Quincke method conventionally used to determine the paramagnetic susceptibility *χ* of solutions works on this assumption. The method assumes that the distribution of paramagnetic ions does not change, and thus, that χ is constant and homogeneous in the solution upon applying a magnetic field. Even if the magnetization force exclusively acts on the paramagnetic ion, they are not supposed to migrate and locally change the solution concentration. As a result, the Quincke calculation of the magnetic susceptibility is a “solid-like” approach. It is performed considering only the weight of the rising liquid as a liquid characteristic: $\chi =\frac{2gh}{\mu_{0}H^{2}}$ where *ρ* is the density, *μ_0_* the permeability of the free space, *g* the gravitational constant, *H* the intensity of the magnetizing field and *h* the height difference with and without magnetic field [41]. The Quincke method has proven its relevance for the determination of the susceptibility of liquids. It is interesting to note that the underlying hypothesis is that of a paramagnetic object and not of a solution whose concentration might vary. This “solid-like” treatment of the susceptibility is coherent with the identification of elastic correlations in paramagnetic solution (Figure 4a,b) and explains the possibility of a collective effect in response to an external field. Recent theoretical developments evidence that liquids exhibit shear elasticity well-above nanoscopic scales. A scale dependence is foreseen [42,43,44]. This is the case of nonaffine models inspired from elastic and viscoelastic constants in glasses of polymers and colloids (NALD approach [45,46,47]). Other authors substitute the description in terms of position and potential of molecular pair by a description in terms of wave vectors and energy densities, which presents the advantage of representing the whole sample characteristics [48,49]. This approach takes into account various experimental results and successfully demonstrates the need to understand the laminar Newtonian regime as the asymptotic limit of the shear elasticity, which is in agreement with the transition observed in Figure 1 and Figure 2. A very recent statistical model proposes a phononic approach in which elastic wave-packets interact with “liquid” particles [50]. Interestingly, this model integrates the recent experimental observation of a possible thermomechanical coupling in liquids [51,52]. The phononic approach is also experimentally corroborated by original neutron studies on self-assembly lamellar structures [53], while new dielectric measurements provide direct experimental evidence on the universal occurrence of extra-slow Arrhenius processes in the liquid dynamic [54]. About thirty years after the pioneering experimental works of Derjaguin and co., in which they interpreted the solid-like liquid behavior at the scale of several microns as an intrinsic liquid property [1], these mesoscopic developments on ionic and paramagnetic liquids reaffirm the importance to consider small-scale liquid flows as nonhydrodynamic.

## 4. Conclusions

We have highlighted that both [emim][Tf2N] and the paramagnetic liquid exhibit a collective response in response to a low-frequency mechanical shear strain; an elastic-like response is observed in nearly static strain conditions at low gap thicknesses (0.100 mm and 0.435 mm). The existence of elastic correlations in a liquid phase away from any phase transition indicates that there is no instant dissipation of the mechanical energy in the thermal fluctuations [51,52,55,56]. Because of the elastic component, the mechanical deformation energy is partly stored. As consequence, the internal energy increases upon applying a low-frequency shear strain. We suspect that a coupling similar to the mechanical stress might occur when applying a magnetic field. Indeed, the structural study carried out using neutron scattering in a wide scattering range is not able to identify either the emergence of a local order that nucleates and grows or a phase separation. In other words, correlations are not induced by an external field but might be pre-existent, which is in agreement with the identification of elastic correlations under mechanical stress and with molecular dynamic simulations, pointing out the need to include the polarizability to reproduce experimental data [25,26,27] and the Quincke’s method that assumes a nonmigration of the paramagnetic ions as an intrinsic hypothesis for the magnetic measurement [41]. In comparison with another imidazolium ionic liquid, increasing the alkyl chain length in ionic liquids seems to reinforce the shear elasticity [57], which is also coherent with the observation of local layering as the chain lengths increase [34,35,36]. The present study provides the evidence that collective nonhydrodynamic behaviors can take place without a long-range structure of the liquid.

## Figures and Tables

**Figure 1 molecules-27-07829-f001:**
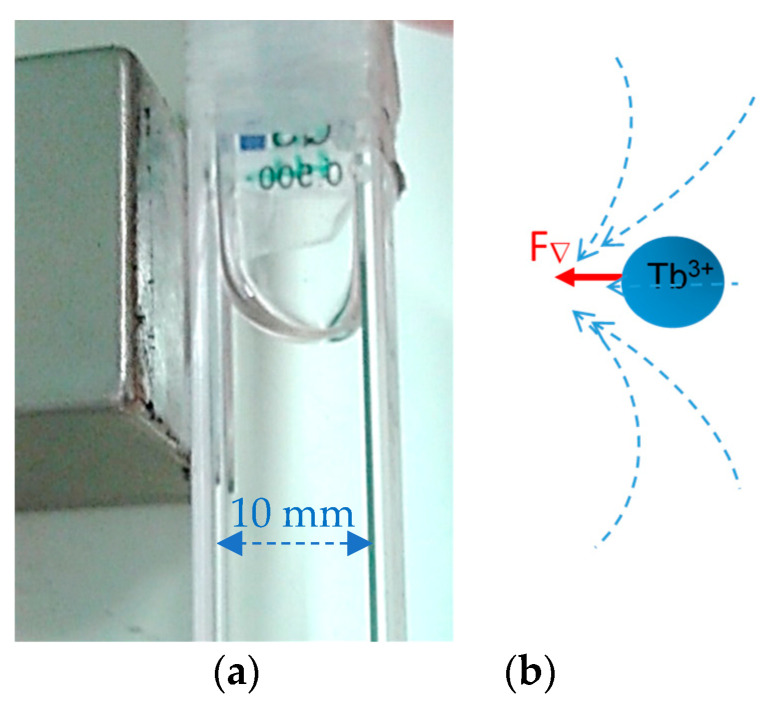
(**a**): Liquid rising in a bottle containing the paramagnetic solution (Tb(NO_3_)_3_, 1M), whose surface is deformed due to the application of an inhomogeneous magnetic field (photograph). The whole solution is drawn to the pole of a 2T magnet. (**b**): The magnetic gradient force (F_∇_) acts on the paramagnetic ions which drain the solution until equilibrating the forces by the weight of the attracted volume.

**Figure 2 molecules-27-07829-f002:**
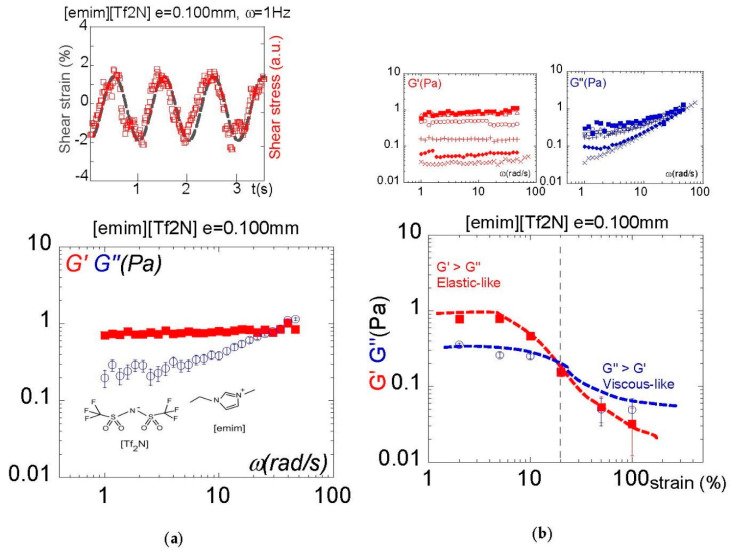
(**a**): Top: in-phase (input) shear strain and (output) shear stress waves at 1 Hz frequency and 2% shear strain by a 100 µm layer of [emim][Tf2N] (1-Ethyl-3-methylimidazolium bis(trifluoromethylsulfonyl)imide). Bottom: The frequency dependence of the viscoelastic moduli indicate a low-frequency elastic-like presence, with the shear modulus *G*′ being higher than the viscous modulus *G*″ up to 20 rad/s. Measurements carried out at 100 µm gap thickness, γ = 2% strain amplitude, room temperature and total wetting conditions (Alumina). (**b**): Top: frequency dependence of the viscoelastic moduli (*G*′, *G*″) displayed by the ionic liquid [emim][Tf2N] at 100 µm gap thickness and different strain amplitudes. Bottom: Shear strain-induced transition from elastic-like to viscous-like. The shear modulus *G*′ higher than the viscous modulus *G*″ progressively vanishes with increasing strain. Measurements carried out at 2 rad/s, at room temperature using total wetting boundary conditions (alumina). The vertical dashed line indicates the separation line between the elastic and the viscous regimes. The continuous lines are eye guides.

**Figure 3 molecules-27-07829-f003:**
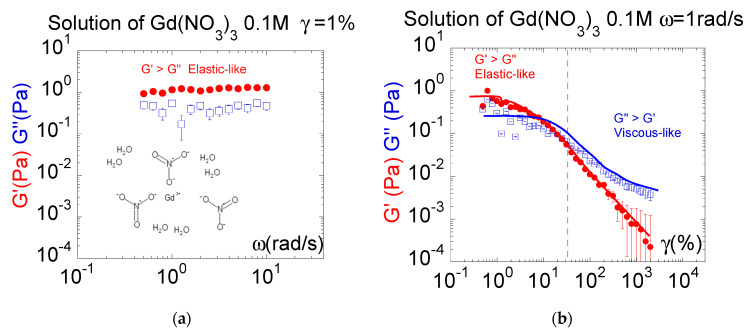
(**a**): Low-frequency elastic response exhibited by an aqueous solution of 0.1 M of Gd(NO_3_)_3_ versus frequency. The shear modulus *G*′ is higher than the viscous modulus *G*″, indicating an elastic-like behavior. Room temperature measurements using total wetting conditions (Alumina substrate), sample thickness: 0.430 mm, shear strain: 1%, logarithmic scale. (**b**): Shear strain behavior displayed by the 0.1 M aqueous solution of Gd(NO3). The shear modulus G′, higher than the viscous modulus G″, vanishes with increasing strain, making the conventional viscous behavior appear. Measurements carried out at 0.430 mm thickness, 2 rad/s, room temperature using total wetting boundary conditions (alumina). The vertical dashed line indicates the transition between elastic and viscous regimes. The continuous lines are eye guides.

**Figure 4 molecules-27-07829-f004:**
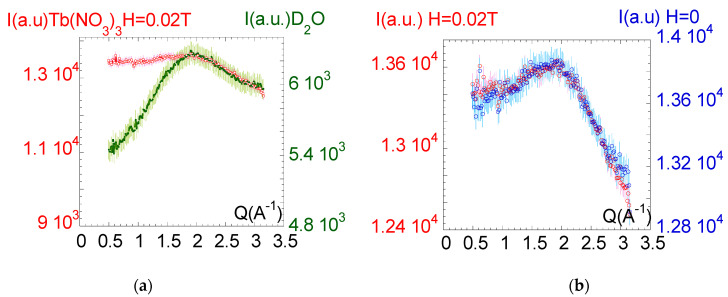
(**a**): Wide-angle neutron diffraction (Barotron G43, *λ* = 2.34 Å, *∆λ/λ* ≅ 1%, *d* = 0.05 m). Heavy paramagnetic atoms are strong radiation absorbers, as the comparison between the D_2_O scattering and the deuterated solution of Tb(NO_3_)_3_ 0.1 M shows. Red points correspond to the paramagnetic solution. Green points correspond to the deuterated water scattering (first peak of the structure factor) and serve as a reference. The error bar is estimated at 5% of the coherent signal. (**b**): Wide-angle neutron diffraction (Barotron [38], *λ* = 2.34 Å, *d* = 0.05 m). Comparison of the scattering curves of the first peak of the structure factor of the deuterated solution of Tb(NO_3_)_3_ 0.1 M with and without magnetic field. The sample is placed in homogeneous magnetic field oriented perpendicular to the incident beam. Blue points correspond to the scattering without magnetic field while red points correspond to the scattering when the magnetic field was applied (0.02T delivered by an electromagnet). The error bar is estimated at 5% of the coherent signal.

**Figure 5 molecules-27-07829-f005:**
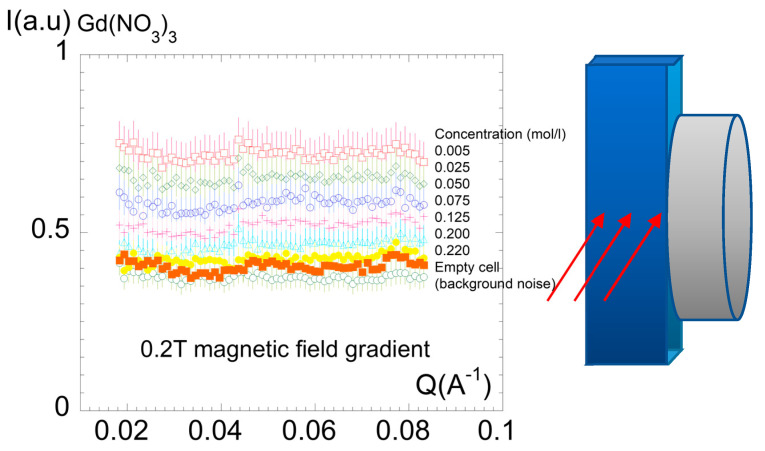
Neutron scattering intensity measured at small-angle scattering on a series of different concentrations of solutions of Gd(NO_3_)_3_ solubilized in deuterated water (PA20 spectrometer). The liquid container (in blue) was placed in contact to a pole of 0.2T magnetic field, as illustrated in the scheme at the right of the figure, to submit the paramagnetic liquid to a field gradient. Room temperature measurements.

## Data Availability

Data supporting the reported results are available on request.

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
