# Peer review of "Collective Effects in Ionic Liquid [emim][Tf2N] and Ionic Paramagnetic Nitrate Solutions without Long-Range Structuring"

_molecules, 2022, doi:10.3390/molecules27227829_

Round 1

Reviewer 1 Report

The authors studied the mechanical properties of ionic liquids and ionic paramagnetic fluids. The results show that the liquids are elastic-like under small strain but become viscous-like under large strain. The authors also found that this change is unlikely to arise from the local ordering of the ionic liquids. The title of the manuscript claims that they identified long-range elastic interactions in ionic liquids.  

The study is timely, and the results look very interesting. Still, I’m afraid that the whole manuscript, including the abstract, is poorly written, and I found it largely unclear although I am familiar with this study topic. For example, it is tough to read the identification of the claimed long-range interactions. Some discussions in the conclusion section should be placed in other sections. I feel strongly that general readers will find the present manuscript unreadable. Thus, although the current results are exciting, I suspect it is not ready for submission yet. I suggest that the manuscript be reviewed by native English speakers to improve readability and transparency. Accordingly, I rated “Overall Merit” as “High”, but I’m afraid that the manuscript should be rejected this time.

Reviewer 2 Report

The paper described the effects of effects of paramagnetic ions over the rheology and structure of solutions and compare with ionic liquids. The subject is interesting, the introduction and methodology are well written, but the discussion and presentation of the results must be improved and the paper should be reviewed again. Specially, the connections between paramagnetic character and the rheology results and even with possible structure formation is not clear to this reviewer and this is supposed to be the main point of this work in my understanding. Other small issues should also be corrected as listed below:

Line 64:

“They are generally characterized by a very low surface tension, non-

flammability, thermal stability and excellent solvating properties.”

It would be good to include some reference(s) here. Also, I would not say that ILs have a “VERY” low surface tension. It is true that most of them have surface tension smaller than water, but not much smaller, with many examples showing surface tension between 40 and 60 dyn/cm, which is larger than the values from organic solvents like ethanol, acetone and alkanes.

Line 85:

“shifting to lower values of scattering vector with in-

creasing chain length n ≥ 5.8, 9, 13.”

These numbers in the end are strange. Maybe 8, 9, 13 should be references?

Line 87:

“The short alkyl chain and the ion symmetry do not allow any particular intermolecular

structuration.”

This phrase need to be clarified, specially what authors mean by “particular intermolecular structuration”, and also what should be the role of the ion symmetry.

Figure 1: The formatting of this figure needs improvement. Some panels don’t have any information of what are the axis and each panel is with a different size. Some have the specification of units as axis labels while others have as inset text... I recommend authors to redo the images to let the formatting more uniform throughout the manuscript, which makes the reading easier.

Line 176: “These measurements are

carried out at relative large gap thickness (430μ m).”

Any justification to why using a different gap thickness? Also, how this could affect the results?

Figure 2: The salt solutions seem to be relatively dilute (0.1 M). It would be interesting if authors can comment if this concentration were expected to produce significant effects over the rheological behavior of the system, in comparison with pure water. Also, it is not clear to this reviewer if and how the magnetic behavior of the cation is affecting the rheology. If a similar Me(NO3)3 salt were used, but with a diamagnetic Me3+ cation instead, this would change anything? I expected this to be one of the main points of the manuscript based on its own abstract and introduction but these results are barely discussed.

Line 193: The strong absorption coefficient of rare earth atoms (Li, Cd, Gd)

Lithium and Cadmium are not rare earth elements.

Line 212: “the q range from 0.01 up to 0.1 Å −1”

Authors used low case q here and upper case in the images. Again, try to keep things uniform for better reading.

Line 226: “The magnetic field (or gradient of magnetic field) does not induce struc-

tural change measurable by scattering methods. This result might find the same explana-

tion as for ionic liquids that require long chain to allow a significant local ordering. Cor-

relatively, magnetically induced collective properties might find some tracks of interpre-

tation related to elastic liquid properties described above (Fig.2).”

Again, the connections are not clear. Which effect in the paramagnetic salt solution would give a similar result as increasing IL alkyl chain? And how magnetic properties are affecting the elastic behavior? These point should be worked out better. Maybe a comparison to a similar diamagnetic salt would be important here.

Reviewer 3 Report

Comments on molecules-1930224:

The current experimental investigation of long-range elastic interactions within the ionic solvent 1-ethyl-3-methylimidazolium bis(trifluoromethylsulfonyl)imide seems interesting to the ‘moleucles’ audience. However, it would be beneficial to perform further experiments to pursue more general insights and improve the scientific quality of the paper. Below, I provide some detailed comments on scientific quality and presentation issues.

I’m not sure why the abbreviation RT is used in the title. All the other words are not abbreviated, and in the abstract the full name ‘room temperature’ is used instead of the abbreviation RT. For consistency and clarification, this RT abbreviation should be written in its full form.

Inconsistent formatting: The figures are cited as Fig. in the main text, but they are presented as Figure in the caption.

No uncertainty estimate is reported for observables, which seems informal and lacks statistical analysis.

Line 175. Subscript of Gd(NO3)3.

Line 224. The data from NIST database requires some citation.

0.1 and 0.01 are presented as 0,1 and 0,01 in many figures. Why?

The authors discuss about the impact of increasing the alkyl chain length and relate it with the observations in the current work. Direct observations would make the work more solid and thus it is beneficial to really do the experiment for species with lengthened aliphatic chains. Based on this chain-length comparison, quantitative results (e.g., regression for some general correlation) could be accumulated.

Molecular simulation as a complementing tool in ionic-liquids research is rather popular in recent years. Although the current paper reports the experimental characterization of many physical properties (e.g., shear strain), it is beneficial to simulate the system to compute these observables and grab further details and insights at atomic level. The current ionic solvent as a well-studied system has been investigated computationally in a number of publications and modelling protocols for accurate descriptions have been summarized (see references such as ChemPhysChem 2007, 8, 2464-2470, Adv. Theory Simul., 2022, 2200274, and J. Phys. Chem. B, 2010, 114, 4984–4997). Following similar protocols, it is quite simple to obtain the computational results for comparison with experiment.  

Round 2

Reviewer 1 Report

The authors largely revised the original manuscript. However, I still strongly feel that the discussion about the magnetic particles has a gap in its logic. First, it’s unclear whether the magnetic field is applied in Figure 2. If applied, what is the magnitude of the field? Secondly, the authors switch to discussing the response of the magnetic particles to the external magnetic field after discussing the elastic moduli in Figure 2. However, it’s unclear how these two things are logically connected. Related to this question, the authors may want to use a subheading to classify the discussion point if necessary.

Reviewer 3 Report

The current version seems fine.

As a last comment, all experimental observations should include some estimates of uncertainty, which has been pointed out in my previous comments but the authors simply overlook it. Although this is not mandatory, the solidity/reliability would be improved if the authors could properly determine these quantities. 
